# Precise Volumetric Measurements of Any Shaped Objects with a Novel Acoustic Volumeter

**DOI:** 10.3390/s20030760

**Published:** 2020-01-30

**Authors:** Viktor A. Sydoruk, Johannes Kochs, Dagmar van Dusschoten, Gregor Huber, Siegfried Jahnke

**Affiliations:** 1Institute of Bio- and Geosciences, IBG-2: Plant Sciences, Forschungszentrum Jülich GmbH, 52428 Jülich, Germany; j.kochs@fz-juelich.de (J.K.); d.van.dusschoten@fz-juelich.de (D.v.D.); g.huber@fz-juelich.de (G.H.); s.jahnke@fz-juelich.de (S.J.); 2Biodiversity, University of Duisburg-Essen, Universitätsstr. 5, 45117 Essen, Germany

**Keywords:** acoustic volumeter, object volume, surface porosity

## Abstract

We introduce a novel technique to measure volumes of any shaped objects based on acoustic components. The focus is on small objects with rough surfaces, such as plant seeds. The method allows measurement of object volumes more than 1000 times smaller than the volume of the sensor chamber with both high precision and high accuracy. The method is fast, noninvasive, and easy to produce and use. The measurement principle is supported by theory, describing the behavior of the measured data for objects of known volumes in a range of 1 to 800 µL. In addition to single-frequency, we present frequency-dependent measurements that provide supplementary information about pores on the surface of a measured object, such as the total volume of pores and, in the case of cylindrical pores, their average radius-to-length ratio. We demonstrate the usefulness of the method for seed phenotyping by measuring the volume of irregularly shaped seeds and showing the ability to “look” under the husk and inside pores, which allows us to assess the true density of seeds.

## 1. Introduction

Determining the volume of an object can be a challenging task. It can be done according to the Archimedes principle by measuring the amount of fluid being displaced. However, such an approach has many drawbacks, particularly when objects are tiny, irregularly shaped, or display rough surfaces, potentially trapping air. To overcome the issue with trapped air, liquids such as toluene [1] or kerosene [2] are sometimes used because of their low surface tension. However, some objects, such as plant seeds, should not be exposed to chemicals, which may stick to the surface, potentially causing large errors in volume calculation and problems in their subsequent use. In this case, noninvasive methods should be used.

One of these methods is optical. Here, projections or stereoscopic images of an object at different angles with subsequent three-dimensional reconstruction allow calculating the object volume [3]. This method can be fast, but requires significant computational resources, as well as multiple cameras or a single camera moving relative to the object. Currently, we are using this method for phenotyping small plant seeds [4,5], but this technique is generally used for objects from micro- to macro-scale [6,7,8]. However, it does not work for all geometries or for objects with rough surfaces [5], since concave structures may not or only partly be visible for cameras.

Another method is gas pycnometry, based on both gas displacement and the thermodynamic law of Boyle–Mariotte [9,10,11]. Though it can be precise and accurate, the method is rather slow, requires either compressed air or inert gas, and temperature control is needed, especially at high pressure [12]. This method is implemented in several commercially available devices, for example, AccuPyc series devices (Micromeritics Instrument Corp., USA) or Ultrapyc and Pentapyc series (Quantachrome Instruments, Anton Paar GmbH, Austria).

A third modality was patented by Mathias [13]: The displacement of air by an oscillating signal applied to a loudspeaker inside a closed system made it possible to compare a known volume with an unknown one. Later on, this device was named the “acoustic pycnometer” [14], which is somehow misleading, since volumes, not densities, were measured. Further development was implemented in several prototypes, such as the acoustic bridge volumeter [15,16], the acoustical capacity/volume meter (RION Co., Ltd., Japan) [17], and their varieties [18,19,20]. There is also another type of acoustic volumeter based on Helmholtz acoustic resonance. In this method, the volume of a cavity defines a resonating sound pitch, which can be determined by a microphone after the excitation of acoustic waves with different frequencies using a loudspeaker at the entrance of the cavity. This technique is used to measure, e.g., tank filling by a liquid [21,22,23] or combustion chamber volumes [24].

Acoustic volumeters have substantial advantages over other methods. They can operate several times faster, do not require large computational resources, and are not limited by object geometry. Acoustic volumeters have few intrinsic limitations. Object size is generally limited only by volumeter size. The detection of closed voids inside the object requires highly sophisticated modalities, such as magnetic resonance imaging [25] or x-ray computed tomography [26], instead of acoustic volumeters. 

However, the cited acoustic volumeters [15,16,17,18,19,20] have several disadvantages. Firstly, the measuring chamber is connected with a closed volume on the back side of the loudspeaker. This closed volume acts as a pressure chamber and, since part of the energy supplied to the loudspeaker is used to compress/expand its volume, leads to a reduction of sensor sensitivity and to a significant non-linearity of the system. Secondly, a connecting tube between the chambers is needed to equalize pressure and humidity. Depending on both frequency and opening size, turbulent flow and redistribution of pressure in the chambers may occur, reducing sensitivity and impairing reproducibility. Moreover, these volumeters were designed to operate at a narrow frequency range and were used predominantly for the measurements of objects with smooth surfaces and volumes larger than 1 mL, such as mass standards [17,20].

Here, we present a novel acoustic volumeter [27], which was developed to overcome the mentioned issues. It can be easily manufactured and used and enables accurate measurements of any shaped objects with volumes from 1 µL to the mL range. We also show that, by operating the acoustic volumeter at different frequencies [27], additional surface information can be obtained, such as surface porosity of an object. The theory behind the new device will be presented, and possible applications will be discussed.

## 2. Materials and Methods

### 2.1. Sensor Design

The overall design of the volumeter is rather simple (Figure 1a): A measured object (*7*) is placed on a flat polished surface (*1*), e.g. a glass plate, and then enclosed by the sensor, whose bottom ending is also polished. No additional force is required for sealing because the measuring chamber (*9*) is already tight enough, due to gravity. The sensor consists of excitation and detection parts. The excitation part consists of a loudspeaker (*5*) (electromechanical transducer [28]) from commercially available earphones (for example, transducers from a Nokia stereo headset WH-102, Nokia Corporation, Finland, or EarPods, Apple, USA, providing a strong central part of their membranes; both types of earphones were used in this work). The detection part can have different designs, which are presented in Appendix A. We selected the inductive head modality due to its simplicity. It is based on the same loudspeaker (*6*) design as the excitation part, which additionally simplifies the construction. Both loudspeakers are tightly fixed inside the sensor housing (*2*) using a separation ring (*4*) and a fixing cap (*3*). The excitation membrane of loudspeaker *5* compresses and expands the air of the measured volume, which is enclosed by the sensor body and surface *1*. Loudspeaker *6* is used as a dynamic microphone, repeating and recording the membrane movements of loudspeaker *5*.

Each loudspeaker has a non-flat frequency response, especially when operating close to its resonant frequencies. This response changes when loading the membrane by placing an object into the chamber. In normal (single frequency) operation mode, the acoustic volumeter is run at a fixed frequency chosen to ensure a flat frequency response for differently-loaded loudspeaker *5*. However, if a frequency sweep is performed, for each frequency, the membrane will compress/expand the air to different pressures, compared to the empty chamber measurements. To avoid the wrong estimation of measured object volumes in such a case, we implemented an additional electret microphone (*8*), which monitors the compression/expansion rate at different frequencies and controls the amplitude of the signal applied to loudspeaker *5*.

### 2.2. Data Acquisition System

The measurement setup is presented in Figure 1b. The data acquisition of the sensor was based on a multifunction data acquisition (DAQ) device (USB-6211, National Instruments, USA). The excitation part was connected to its digital-to-analog converter through a homemade voltage-to-current signal converter (VCC; based on UA741, Texas Instruments, USA, and BD139/BD140, STMicroelectronics, Switzerland). The signal from the detection part was amplified using a homemade operational amplifier (OA; based on LM358AP, Texas Instruments, USA) and recorded by an analog-to-digital converter of USB-6211. Instead, a common soundcard can be used to make a tiny device wirelessly linked to a smartphone or a computer. To control the multifunction DAQ device, the LabView- (National Instruments, USA) based program was developed and used.

## 3. Working Principle

During measurement, a sinusoidal current, $I=I_{0}\sin\left(2\pi ft\right)$, flows through the coil of the loudspeaker *5*. The coil moves the excitation membrane up and down due to the Lorenz force, *F*, which can be written as [28]:(1)$$F=B_{1}l_{1}I=B_{1}l_{1}I_{0}\sin\left(2\pi ft\right),$$
where *B*_1_ is the magnetic flux density, *l*_1_ is the length of the wire in the coil of the loudspeaker *5* (identical to microphone *6* in our application), *I*_0_ is the amplitude of the applied current, *f* is the frequency, and *t* is the time. The force will be distributed into two parts, one of which concerns losses, *F_loss_*, such as bending of the membrane or overcoming inertia [28,29]. The second major part, *F*-*F_loss_*, expresses the compression and expansion of the air inside the sensor. Since the membranes of both transducers, *5* and *6*, are very light and the forces needed to bend them are very small, we may neglect *F_loss_* in a first approximation, and thus force *F* describes compression of the air inside the sensor with the following change in pressure:(2)$$\Delta p=ℛ\left(f\right)\frac{F-F_{loss}}{S}\approx \frac{F}{S},$$
where *S* is the area of the membrane and ℛ(f) is the frequency response of the loudspeaker, as mentioned in Section 2.1.

The sensor chamber is not a closed system in terms of heat exchange. At low frequencies (here, <0.1 Hz), an isothermal state is established, i.e., the air can equalize its temperature at each position inside the measuring chamber with the wall temperature of the sensor. At high frequencies (here, >10 Hz), thermal equilibrium cannot be established, and the sensor rather works with adiabatic compression and expansion (more details in Appendix B). We can thus consider a polytropic regime [30] over the whole frequency range of the sensor operation, where the polytropic index, *γ_P_*, varies from 1 (f→0) to ~1.4 (f→∞; the adiabatic index of air). Such a regime can be described as follows:(3)$$p_{a}V_{0}{}^{\gamma_{P}}=\left(p_{a}+\Delta p\right){\left(V_{0}-\Delta V\right)}^{\gamma_{P}},$$
where *p_a_* is the atmospheric pressure, *V*_0_ is the initial volume of the sensor, and ∆*V* is the volumetric change due to the movements of the loudspeaker *5* membrane. Assuming that the change in pressure ∆*p* is very small, relative to *p_a_* (see estimation in (29)), the volume change ∆*V* can be approximated by:(4)$$\Delta V\approx \frac{1}{\gamma_{P}}\Delta p\frac{V_{0}}{p_{a}}.$$

Microphone *6* duplicates the motion of the loudspeaker *5* due to pressure-wise coupling and generates an electromotive force, ε, following [28],
(5)$$\varepsilon =B_{2}l_{2}v=B_{2}l_{2}\frac{1}{S}\frac{\partial \Delta V}{\partial t},$$
where *B*_2_ is the magnetic flux density inside the coil of the microphone *6*, *l*_2_ is the length of the wire in the coil, and *v* is the velocity of the coil displacement. Combining Equations (1), (2), (4), and (5) one obtains
(6)$$\varepsilon =\frac{2\pi fl_{1}l_{2}B_{1}B_{2}I_{0}\cos\left(2\pi ft\right)ℛ\left(f\right)}{S^{2}p_{a}\gamma_{P}}V_{0}.$$

For our purposes, a suitable measure of ε is its root mean square (rms) value:(7)$$\varepsilon_{rms}=\frac{2\pi fl_{1}l_{2}B_{1}B_{2}I_{0}ℛ\left(f\right)}{\sqrt{2}S^{2}p_{a}\gamma_{P}}V_{0}.$$

When we put an object with volume *V_S_* inside the sensor, the initial volume *V*_0_ decreases to *V*_0_ − *V_S_*. Applying the same current *I* to the loudspeaker *5* as before, the electromotive force induced in the coil of microphone *6* is reduced due to the presence of the object:(8)$$\varepsilon_{rms}^{\left(s\right)}=\frac{2\pi fl_{1}l_{2}B_{1}B_{2}I_{0}{ℛ}_{S}\left(f\right)}{\sqrt{2}S^{2}p_{a}\gamma_{P}}\left(V_{0}-V_{S}\right),$$
where ${ℛ}_{S}\left(f\right)$ is the frequency response of the loudspeaker when the object is present inside the chamber. When we assume $ℛ\left(f\right)={ℛ}_{S}\left(f\right)$, which holds in case of operating far from internal resonances of the loudspeaker *5*, the difference between $\varepsilon_{rms}$ and $\varepsilon_{rms}^{\left(s\right)}$ relative to $\varepsilon_{rms}$ is a simple relation:(9)$$\varepsilon_{rel}=\frac{\varepsilon_{rms}-\varepsilon_{rms}^{\left(s\right)}}{\varepsilon_{rms}}=\frac{V_{S}}{V_{0}},\;\mathrm{or}$$
(10)$$V_{S}=\varepsilon_{rel}V_{0}.$$

Thus, object volumes *V_S_* can be calculated from known *V*_0_, acquiring $\varepsilon_{rms}$ for an empty chamber each time before measuring $\varepsilon_{rms}^{\left(s\right)}$.

It should be noted that by working at frequencies where the assumption $ℛ\left(f\right)={ℛ}_{S}\left(f\right)$ is questionable, an additional microphone, here *8*, should be used to control the frequency response of the loudspeaker *5* by correcting the applied signal.

## 4. Results and Discussion

### 4.1. Single Frequency Operation

#### 4.1.1. Measurement of Objects with Known Volumes; Calibration Curves

As proof of principle and to demonstrate both precision and accuracy, we measured stainless steel balls with different diameters of 1.000, 1.588, 2.500, 4.000, 5.000, 6.000, 7.000, 8.000, and 9.000 (mm; ±0.011 mm) and various combinations thereof (Figure 2). For the sensor shown in Figure 1, *V*_0_ was about 1300 µL. We added different parts to the sensor to vary its volume. A conical insert reduced the volume to ~600 µL and a 10 mm-long cylindrical extension increased the volume to ~2630 µL. A frequency of 232 Hz was chosen to operate in the adiabatic regime (>10 Hz) to make measurements fast and to be far enough from the resonance frequencies of the loudspeakers (~800 Hz), the network frequency (50 Hz), and its harmonics. Preferably, the working frequency should not exceed a frequency with a corresponding sound wavelength less than four times larger than the largest distance within the chamber. This will avoid uncertainties caused by acoustic effects, such as standing waves, diffraction, etc.

The measured data (Figure 2) demonstrate linear behavior. By fitting with Equation (9), the sensor volumes *V*_0_ can be calculated and compared with geometrically estimated values, see Table 1. The values obtained using Equation (9) are higher than geometrically estimated, which can be explained by losses in the transducer membranes, i.e., *F_loss_* in Equation (2). The smaller the measured objects, the larger is the membrane distance moved by the same current amplitude, *I*_0_, and therefore *F_loss_* becomes higher, which, in Equation (9), reflects an increase of *V*_0_. Taking this effect into account (see Appendix C), more precise relations than Equations (9) and (10) can be obtained by:(11)$$\varepsilon_{rel}=\frac{V_{S}}{V_{0}\left(1+C\left(V_{0}-V_{S}\right)\right)},\;\mathrm{or}$$
(12)$$V_{S}=V_{0}\varepsilon_{rel}\frac{1+CV_{0}}{1+CV_{0}\varepsilon_{rel}},$$
where $C=K_{ms}ℛ\left(f\right)/\left(\gamma_{P}p_{a}S^{2}\right)$ is another fit parameter and $K_{ms}$ is the stiffness of the mechanical suspension of loudspeaker *5* and microphone *6* [28,29]. Using Equation (11), the fits became more accurate (i.e., the reduced chi-squared, $\chi_{\nu }^{2}$, values (see Table 1) became closer to 1 and the dashed lines in Figure 2 passed through more measured points) and the determined volumes, *V*_0_, almost matched the geometrically estimated volumes. By calibrating the sensor using volumes of known objects, e.g., metallic balls, and measuring objects with *V_S_* at least 10 times smaller than *V*_0_, Equation (10) delivered an accuracy of < *V*_0_ /500 and a precision of < *V*_0_ /2000 for *V_S_* estimation, whereas with (12), the accuracy easily reached *V*_0_/1000 over the entire range of *V_S_* measured in this experiment and a precision of < *V*_0_ /2000 for *V_S_* ≤ *V*_0_/50 and about 2.5% of *V_S_* for *V_S_* > *V*_0_/50.

The sensor was quick to operate: less than 5 s of measurements at 232 Hz gave plenty of sinusoidal signal periods to estimate the object’s volume with the described accuracy and precision.

#### 4.1.2. Measurement of Plant Seeds

The sensor is suitable for measuring objects of any shape. To demonstrate its practical feasibility, we measured plant seeds of barley (*Hordeum vulgare* Barke) and wheat (*Triticum aestivum* Scirocco). Obtaining $\varepsilon_{rel}$ for each seed and using Equation (12) with the known *V*_0_ and *C* from the calibration procedure, we calculated seed volumes *V_S_*, which were compared with respective masses obtained by a balance (Figure 3). The linear fitting of the two data sets (solid lines) resulted in densities of 1.45 ± 0.03 g mL^−1^ and 1.47 ± 0.04 g mL^−1^ for barley and wheat seeds, respectively. These values were larger than data obtained by optical methods, e.g., shooting projections of seeds at different angles and further constructing a three-dimensional model for each individual seed [4], giving densities of 1.25 ± 0.06 g mL^−1^ [5] and 1.34 ± 0.03 g mL^−1^ (data not published yet) for barley and wheat seeds, respectively. In the case of, e.g., barley seeds, the surface is rather rough leading to volume overestimation by the optical methods, resulting in underestimation of seed densities [5]. With the acoustic volumeter, compressed air can penetrate inside tiny pores/holes of the surfaces, making measurements almost independent of surface roughness or irregular seed shape, which allows more precise volume estimation. However, pores on the surface of an object can lead to different volumes when measured at different frequencies, *f*, as described in the next section.

### 4.2. Frequency Sweep Operation

#### 4.2.1. Measurements of Total Volume of Pores

Here, we suppose that one is measuring an object that has pores with radius *r_p_* and depth *L_p_* (see Figure 4), resulting in a total volume of pores $V_{Pt}=N_{p}L_{p}\pi r_{p}^{2}$, where *N_p_* is the number of pores. Compressing/expanding air inside the sensor chamber induces an air flow at the entrance of each pore. Due to this flow, the air inside the pores is also redistributed and compressed/expanded. These changes are different at different operating frequencies, *f*, and, because of the finite air viscosity, the pressure changes stops reaching the whole depth of the pores at high frequencies.

To understand in detail how pressure and velocity are distributed inside the pores, we need to solve continuity, Navier–Stokes, and energy equations for a compressible Newtonian fluid [31]:(13)$$\frac{\partial \rho_{p}}{\partial t}+\nabla \cdot \left(\rho_{p}\vec{v_{p}}\right)=0,$$
(14)$$\rho_{p}\left(\frac{\partial \vec{v_{p}}}{\partial t}+\vec{v_{p}}\cdot \nabla \vec{v_{p}}\right)=-\nabla p_{p}+\nabla \cdot \left(\mu \left(\nabla \vec{v_{p}}+{\left(\nabla \vec{v_{p}}\right)}^{T}\right)-\frac{2}{3}\mu \left(\nabla \cdot \;\vec{v_{p}}\right)Ι\right),$$
(15)$$\rho_{p}c_{P}\left(\frac{\partial T}{\partial t}+\vec{v_{p}}\cdot \nabla T\right)=\nabla \cdot \left(\kappa \nabla T\right)+\beta T\left(\frac{\partial p_{p}}{\partial t}+\vec{v_{p}}\cdot \nabla p_{p}\right)+\left(\mu \left(\nabla \vec{v_{p}}+{\left(\nabla \vec{v_{p}}\right)}^{T}\right)-\frac{2}{3}\mu \left(\nabla \cdot \;\vec{v_{p}}\right)Ι\right):\nabla \vec{v_{p}},$$
where *ρ_p_* is the density of air inside a pore, $\vec{v_{p}}$ is the velocity vector, *p_p_* is the pressure, *µ* is the air dynamic viscosity, **I** is the unit tensor, *c_P_* is the specific heat capacity of air at constant pressure, *T* is the temperature, *κ* is the thermal conductivity of air, and *β* is the thermal expansion coefficient of air. These equations should be solved considering both non-slip conditions at the walls (meaning that air flow velocity is zero close to walls) and the changes of pressure at the entrance of a pore obtained from Equation (2). Supposing an isothermal process inside the pores due to short distances to the walls (see also calculations in Appendix B), Equation (15) can be replaced by:(16)$$\frac{p_{p}}{\rho_{p}}=constant.$$

The solving of Equations (13)–(16) analytically is impossible without making critical assumptions. Therefore, we used COMSOL Multiphysics 5.3 with the computational fluid dynamics (CFD) module (COMSOL, Inc, MA, USA) to simulate pressure changes in cylindrical pores of various radii and lengths over several periods of the excitation oscillation. For the purpose of the simulations, the object volume was set to zero, *V_S_* = 0, and the pores were simulated as cylindrical extensions of the chamber volume, *V*_0_. Laminar isothermal flow, as well as two-dimensional axisymmetric analysis, were selected to reduce the simulation time. The time domain simulations were performed using a triangular type of mesh automatically generated by COMSOL. Mesh size depended on the length and radius of the simulated tube, as well as on the chamber volume, resulting in a number of elements ranging from 3000 to 20,000. We performed a mesh convergence analysis for several simulated structures to confirm the mesh size selection.

The detection part of the sensor is measuring the fluctuations of the chamber volume over several oscillation periods during its compression/expansion by the excitation part. Due to the linear pressure dependence on density, as shown in Equation (16), these fluctuations can be represented as pressure changes $\Delta p^{\left(air\right)}\left(\vec{x},\;t\right)$ integrated over the whole simulated volume $\;V_{air}=V_{0}+V_{Pt}$
$\;\left(\vec{x}\in V_{air}\right)$ and divided by *V_air_*, giving the mean pressure changes $\;\Delta p_{mean}^{\left(air\right)}\left(t\right)$ at each time point *t*:(17)$$\Delta p_{mean}^{\left(air\right)}\left(t\right)=\frac{1}{V_{air}}{∭}_{V_{air}}\Delta p^{\left(air\right)}\left(\vec{x},\;t\right)d\vec{x}.$$
The rms-value of $\;\Delta p_{mean}^{\left(air\right)}\left(t\right)$ divided by the rms-value of the applied pressure Δ*p*, as shown in Equation (2), allowed us to estimate the visibility *ξ* of the total simulated volume of air *V_air_*, in relative terms (between 0 (completely invisible) to 1 (completely visible)). Multiplying the visibility by *V_air_* and subtracting the air volume without pores, *V*_0_ gave us the visible volume of the pores $\;V_{Pt}^{\left(visible\right)}$:(18)$$V_{Pt}^{\left(visible\right)}=\xi V_{air}-V_{0}=\xi V_{Pt}-\left(1-\xi \right)V_{0},$$
which depends on the frequency, *f*, of the pressure changes produced by the excitation membrane. 

By sweeping the measurement frequency, we can change the visibility of pores of a measured object. This should allow us to estimate the total volume of the pores and the surface porosity of the measured object. However, internal porosity of the object is not accessible with this approach and would require tomographic methods [25,26].

#### 4.2.2. Teflon Tube Example; Theory Verification

To prove the above-mentioned idea and to compare the simulated results with a real experiment, we measured the inner volume of Teflon tubes (0.5 mm inner diameter) as a proxy of pores. The sensor with an inner volume of 6.5 mm radius and 3 mm length was used for these measurements. The Teflon tubes were closed on one side and opened on the other, with different lengths of 50, 100, 150, 200, 250, 300, and 350 mm. The open end of the tubes was tightly connected to the chamber at the Table 1 side. Their inner volume was measured at a range of frequencies from 20 to 2760 Hz and compared to results from COMSOL simulations (Figure 5a, where only a subset of Teflon tube lengths is shown for clarity). It should be noted that for this comparison, the volume of the sensor chamber was also simulated, which may lead to different results than a simulation of the tubes only, see below. The frequencies between 700 and 1000 Hz were skipped during the measurements due to possible instabilities of the sensor operation close to its main resonance frequency of ~800 Hz (see Appendix D).

Figure 5a shows that at small operation frequencies, the whole internal volume of the tubes was measured. When the frequency increased, the pressure changed faster at the entrance of the tube, resulting in a higher-pressure gradient and, thereof, inflow speed. This in turn caused a higher pressure drop into the tube due to the friction with walls and between air layers inside the tube. At high frequencies, the pressure changes did not reach the back of the tube anymore, so that part of the inner volume became invisible. At a transition frequency, *f*_0_, half of the tube volume was visible for the sensor, and at frequencies much higher than *f*_0_, the whole tube became invisible.

Overall, measured and simulated results in Figure 5a matched well. Yet there was a small (about 1.6%) shift between the simulated *f*_0_^(*s*)^ and measured *f*_0_^(*m*)^ transition frequencies to lower frequencies in the simulations (see Figure 5b). This can be explained by both the uncertainties during the preparation of tubes and increasing of the speed of sound due to the humidity of air [32] (the tubes were measured at RH ≈ 30% and *T* = 27 °C), which was not taken into account during the simulation. We found that the transition frequency had linear dependence on the speed of sound, as shown next.

#### 4.2.3. Influence of the Internal Volume of the Sensor and Pores Geometry

An additional factor, with which the transition frequency may be shifted to lower frequencies, is a phase difference in pressure changes inside the pores and those in the whole sensor chamber. The dependence of this shift on the internal volume of the sensor chamber was calculated by simulation of various volumes of tubes/pores, as well as different chamber volumes of the sensor with different ratios between the radius and length for both of the volumes. The obtained data (see Figure 6) show a simple dependence of the transition frequency, *f*_0_, resulting from simulations of the sensor with internal volume, *V*_0_, on the real transition frequency, *f*_0_^(*real*)^, obtained from simulations of pores only, and on the ratio of pore volume *V_Pt_* and *V*_0_:(19)$$f_{0}=f_{0}^{\left(real\right)}\frac{1+\left(0.631\pm 0.003\right)\left(2.7\pm 0.1\right)V_{0}/V_{Pt}}{1+\left(2.7\pm 0.1\right)V_{0}/V_{Pt}},$$
where coefficients are presented in the form of (value ± standard error). In almost all cases of seed measurements, the pore volumes *V_Pt_* are more than 10 times smaller than the internal volume, *V*_0_, of the sensor, so that measured *f*_0_ ≈ 0.63 *f*_0_^(*real*)^. This can be considered an advantage for the sensor, because it implies that the sensor can detect *f*_0_^(*real*)^ values that are about 1.5 times higher than its maximum working frequency. The simulated data in Figure 6 contain different amplitudes of the pressure changes (1 Pa light gray up triangles, 100 Pa black dots, and 10,000 Pa dark gray down triangles) that can be produced by the excitation membrane, which proves the transition frequency independence of this amplitude.

The transition frequency, *f*_0_, also depends on the ratio between the pore/tube radius, *r_p_*, and its length *L_p_*, as shown in Figure 7. There are two important regions with quadratic and linear dependence, respectively. The quadratic dependence is related to very long tubes/pores, where there is a fully developed parabolic velocity profile of the air flow inside the tube, which was formed at distances much smaller than the length of the tube (see red dashed line in Figure 7a). In this region, the transition frequency *f*_0_ is proportional to the relation *r_p_*^2^/*L_p_*^2^, and in our case, it turned out to be close to the following expression:(20)$$f_{0}=\frac{p_{a}}{10\mu }\frac{r_{p}^{2}}{L_{p}^{2}}.$$
The second region has a close to linear dependence on *r_p_*/*L_p_* and it is related to tubes/pores, where the flow velocity profile is not completely formed within the whole length of the tube (see blue dash-dotted line in Figure 7a). Plotting the data in a different coordinate system (see Figure 7b), it becomes visible that *f*_0_ has a very weak overall dependence on *r_p_* in this region and with decreasing *L_p_*, *f*_0_ asymptotically approaches the following equation (see Appendix E):(21)$$f_{0}\to \frac{\varsigma }{2\pi }\frac{v_{S}}{L_{p}},$$
where *v_S_* is the speed of sound and *ς* = 1.89549 is the solution of sinc (*ς*) = 1/2. Such a law can be obtained considering the sum of the sound pressure waves incident and reflected from the back wall of the pores, assuming that there is no friction between air layers and walls. This provides an additional possibility to measure the average length *L_p_* of all pores. Moreover, knowing *L_p_* and the total volume of pores *V_Pt_*, one can calculate the surface covered by all pores, *N_p_πr_p_*^2^ = *V_Pt_*/*L_p_*. By combining the volumeter with an optical three-dimensional reconstruction method, which can estimate the whole surface area of the measured object, one can obtain the related part of the surface area covered by pores, which may be an important parameter in different fields, such as geology [33], material science, or medicine [34,35].

#### 4.2.4. Plant Seed Example; Under the Husk

To demonstrate how the seed volume measurements depend on frequency, we measured 24 wheat seeds (*Triticum aestivum* Scirocco) and 24 rice seeds with husks (*Oryza sativa* Maratelli) (Figure 8). The measurements were performed with seeds at different moisture levels, i.e., 11%, 8.2%, 1.5%, and 0% of water content (WC). The mean wheat seed volumes were almost independent of the frequency within the error of measurements (here about 1 µL), indicating that no pores on the seed surface became visible in the applied frequency range. On the contrary, rice seeds with husks showed an increasing measured volume with increasing frequency. This demonstrates that at low frequency, the air penetrated more easily under the husk than at higher frequency. These findings are further supported by the measurements at different moisture levels. Seeds normally increase in volume at higher water content [1,36]. Nevertheless, the overall volume (at high frequency) of the rice seeds with the husk remained the same, which is a good indicator of non-swelling husk material.

Comparing the mean volumes of rice seeds with husks for WC = 0% at high-frequency, e.g., ~21 µL at ~3000 Hz, and low-frequency, e.g., ~16 µL at ~150 Hz, we could estimate the mean distance between the husk and the seed. The space under the husk filled a volume of about 21 – 16 = 5 µL, and the surface area, in this case, could be estimated as ~45.5 mm^2^ by representing the rice seed as an ellipsoid with axes of 3.5, 1.5, and 1 mm. Dividing the volume by the surface area gave an estimate of the distance between the rice seed and its husk of about 0.1 mm. 

The measurements at different frequencies demonstrate that the selection of a narrow frequency operation range may lead to inaccurate volume estimation when using an acoustic volumeter. For objects with pores on the surface, a measurement at higher frequencies may result in volume overestimation, since pores might be not visible then. Ideally, only working at infinitely low frequencies (such as gas pycnometry) will allow us to obtain the ‘true’ volume of an object. In practice, measurement at low frequencies, e.g., around 232 Hz, may provide already quite reliable results for, e.g., plant seeds. 

## 5. Conclusions

We presented a simple acoustic volumeter, described its measuring principle analytically, and validated it by measuring objects of known and well-defined size. The system is applicable to objects of any shape and has some novel features. (1) It has a large range of operation, starting at volumes of around 1 µL up to several mL, which can be further scaled up or scaled down. (2) The sensor is rather simple to construct, consisting of two tubes, two electromechanical transducers, and some electronics. (3) The acoustic volumeter is easy to operate. A test object is put on a flat plate and then enclosed by pulling the sensor over the plate. (4) Volume detection is fast. A few seconds are needed to gather enough signal points to estimate object volumes more than 1000 times smaller than the volume of the sensor chamber with both high precision and high accuracy. (5) The acoustic volumeter can be operated at different frequencies between, e.g., ~20 and ~3000 Hz. This allows us to estimate the ‘true’ volume of an object with a rough surface and to assess the total volume of pores on the surface of an object, as well as the relation of the pore dimensions *r_p_*/*L_p_*. When plant seeds were investigated, different volumes measured at different frequencies indicated porous surfaces. This can be used to gather seed properties, which were so far not approachable.

## Figures and Tables

**Figure 1 sensors-20-00760-f001:**
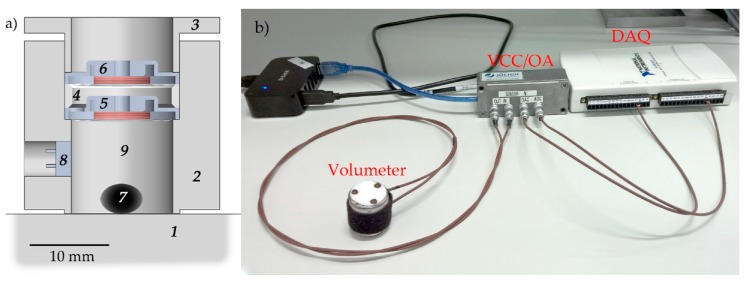
Schematic and image of the acoustic volumeter. (**a**) Schematic cross-section of the acoustic volumeter, which is placed on a plate *1*. *2* cylindrical housing of the sensor; *3* fixing cap; *4* washer/separation ring; *5* loudspeaker with excitation membrane; *6* dynamic microphone with detection membrane; *7* measured object; *8* additional electret microphone to compensate nonlinearities in case of measurements at different frequencies; *9* measuring chamber. (**b**) Measurement setup. The volumeter is connected to a voltage-to-current converter (VCC) and an operational amplifier (OA), which are connected to a multifunction data acquisition (DAQ) system.

**Figure 2 sensors-20-00760-f002:**
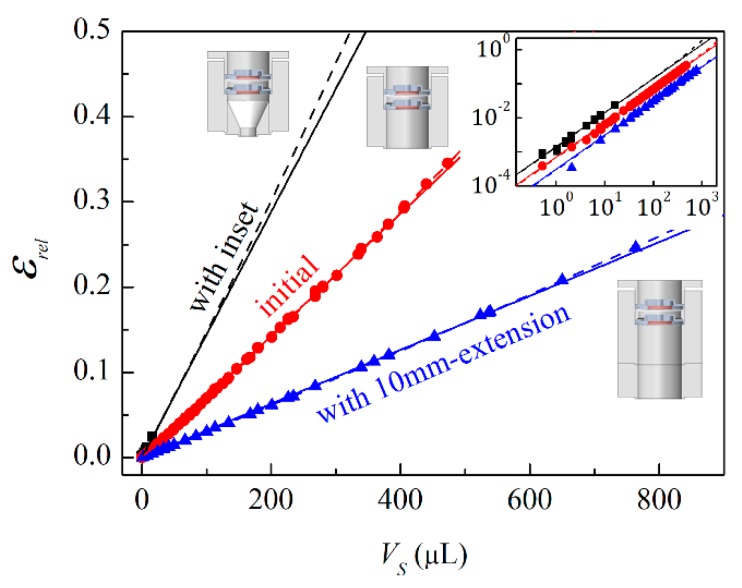
Measurements of stainless steel balls with different diameters and combinations thereof. Symbols reflect measured signal, $\varepsilon_{rel}$, versus the geometrically calculated ball volumes, *V_S_*. Data for the initial sensor (*V*_0_~1300 µL) are highlighted by red circles. Black rectangles and blue up triangles represent data for sensors with a conical inset (–700 µL) and with a 10 mm-extension (+1330 µL), respectively. Solid and dashed lines show the fit results using (9) and (11), respectively. *f* = 232 Hz. Inset: The same data shown in log-log scale.

**Figure 3 sensors-20-00760-f003:**
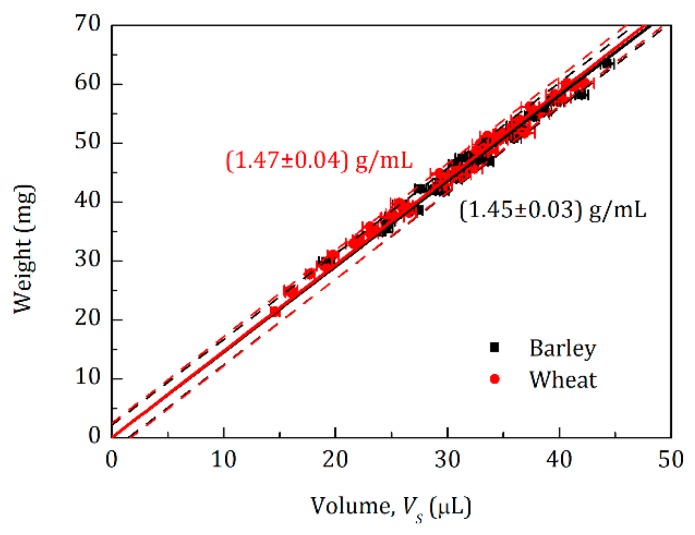
Measurements of 24 barley and 24 wheat seeds. The seeds were weighted using a balance with a precision of 0.1 µg. Volumes were measured by the initial sensor at 232 Hz using the corresponding calibration curve depicted in Figure 2. Right-black and left-red numbers (value ± standard deviation), representing the densities for barley and wheat seeds, respectively, were obtained by a linear fit with zero intercept applied to the black and red data points, respectively. Dashed lines reflect 95% prediction bands for both of the sets.

**Figure 4 sensors-20-00760-f004:**
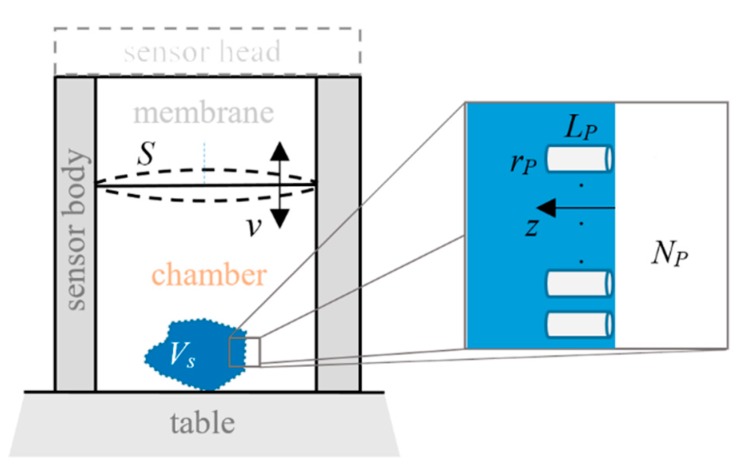
Measurement of an object (schematic view) with volume *V_S_* and with *N_p_* pores on its surface. *L_p_* and *r_p_* reflect the pores’ lengths and radius, respectively. *v* is the speed of the excitation membrane with surface *S*.

**Figure 5 sensors-20-00760-f005:**
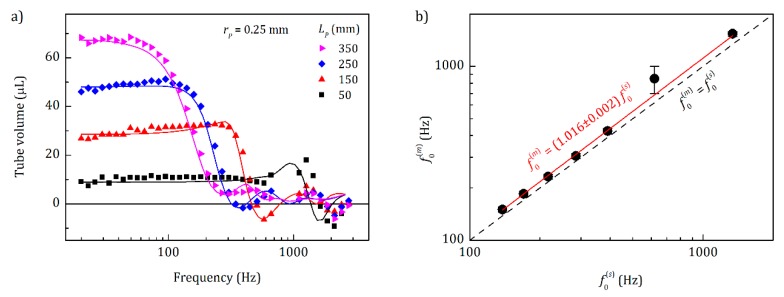
Comparison of measured and simulated results. (**a**) Internal volume of a Teflon tube closed at one side, measured at different frequencies. The inner radius of the tube was *r_p_* = 0.25 mm and its length *L_p_* varied from 50 to 350 mm. Solid lines reflect simulated results using COMSOL Multiphysics with CFD module. (**b**) Measured transition frequency versus simulated transition frequency. The black dashed line reflects a 1:1 dependence (right-black formula) and the red solid line is a linear fitting with fixed intercept resulting in the left-red formula.

**Figure 6 sensors-20-00760-f006:**
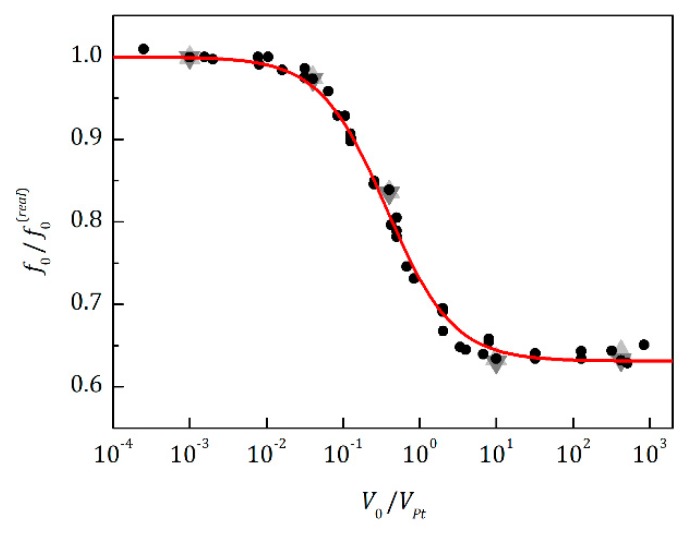
Relation between the transition frequencies, *f*_0_ and *f*_0_^(*real*)^, obtained by pore simulations with (*f*_0_) and without (*f*_0_^(*real*)^) the sensor chamber, in dependence of different pore/tube volumes, *V_Pt_*, and different chamber volumes, *V*_0_. Simulations also contain different pressure amplitudes produced by the excitation membrane (1 Pa, 100 Pa, and 10,000 Pa, depicted as light gray up triangles, black dots, and dark gray down triangles, respectively). Red line shows a fit using *y* = (1 + *abx*)/(1 + *bx*), where *a* and *b* are the fitted coefficients and *y* and *x* are the ordinate and abscissa, respectively.

**Figure 7 sensors-20-00760-f007:**
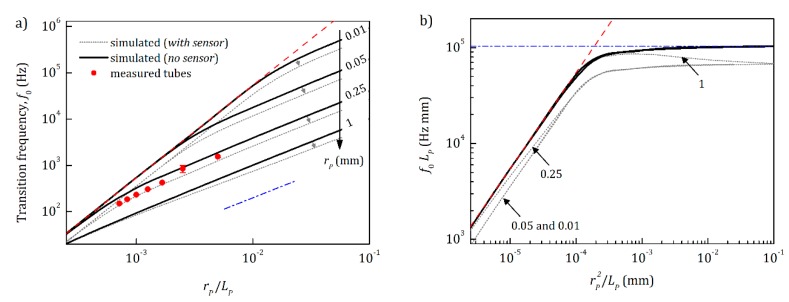
Transition frequency *f*_0_ dependence on the relation between pore/tube radius *r_p_* and length *L_p_*. (**a**) Simulations of a pore/tube without (black solid lines) and with the sensor volume of 6.5 mm radius and 3 mm length (dotted gray lines). Red dots are measured data for the Teflon tube of 0.25 mm inner radius and of different lengths, partially presented in Figure 5a. The shift of *f*_0_ to high frequencies in comparison to theoretically estimated values (dotted gray lines) is discussed in Figure 5b. Red dashed and blue dash-dotted lines show a quadratic $\frac{p_{a}}{10\mu }\frac{r_{p}^{2}}{L_{p}^{2}}$, and linear $\frac{\varsigma }{2\pi }\frac{v_{S}}{L_{p}}$ dependence, respectively. (**b**) The same data represented in different coordinates.

**Figure 8 sensors-20-00760-f008:**
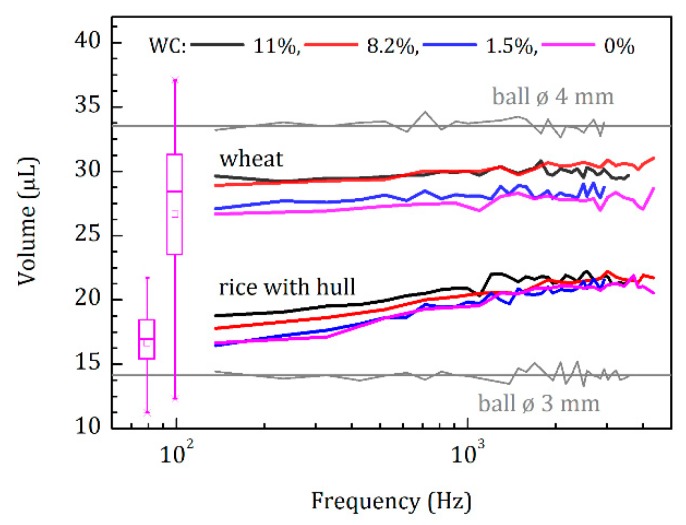
Measurements of wheat and rice seeds at different frequencies. Mean measured volume of 24 seeds versus frequency at 11% (black), 8.2% (red), 1.5% (blue), and 0% (magenta) moisture levels, i.e., water content (WC) by mass. Gray curves and horizontal lines show the measured and calculated volumes of polyoxymethylene (POM) calibration balls with known diameters, respectively. Magenta box charts reflect the variabilities of the respective measured values at the lowest frequency and at WC = 0%.

**Table 1 sensors-20-00760-t001:** Comparison of geometrically estimated initial volumes of the sensor and measured volumes calculated by Equations (9) and (11).

Geometrically Estimated	Fitted by Equation (9)		Fitted by Equation (11)		
*V*_0_ (µL)	*V*_0_ (µL)	$\chi_{\nu }^{2}$	*V*_0_ (µL)	*C* (µL^−1^), ×10^−6^	$\chi_{\nu }^{2}$
600 ± 70 ^1^	693 ± 5	1.10	600 ^2^	260 ± 20 ^2^	1.05 ^2^
1300 ± 70 ^1^	1397 ± 3	19.3	1250 ± 20	130 ± 20	9.66
2630 ± 70 ^1^	3160 ± 10	31.4	2570 ± 30	112 ± 9	3.47

^1^ Errors were estimated taking into account the roughness of the excitation membrane. ^2^ Fitting with fixed *V*_0_ because of too small values for *V_S_*.

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
