# Peer review of "Precise Volumetric Measurements of Any Shaped Objects with a Novel Acoustic Volumeter"

_sensors, 2020, doi:10.3390/s20030760_

Round 1

Reviewer 1 Report

The authors present an interesting bit of work. I like it generally but I have some comments and I would like a reply to each one of them before I can suggest it for publication. 

The authors should discuss optical methods (e.g. photogrammetry) for volumetric measurements. Line 109 – Can the authors explain the validity of the assumption that the frequency response is flat. This is an important aspect as many of the presented results show a frequency dependence which may be accounted for with the speaker response. A FRF of the transducer used in the study should be presented in the appendix. Pictures of the experimental set-up should be included. Line 138 – The same number of significant figures should be used. Figure 2. The inserted diagrams of different set-ups should be larger. There is enough room. Equation 12. At the right hand side there is a corrupted symbol. Equation 14. Same as above. Line 219. Some details of the FEA implementation should be added (in an appendix possibly). Mesh type, element size, time domain or frequency domain etc. Corrupted symbols. These are all throughout the paper and all of them need to be addressed. Line 243. Why is the frequency range to specific? More detail is needed on exactly which devices were used for both the transmitting and receiving signals. Line 246 talks about resonance effects of the system but no data is presented to confirm this. This data needs to be included here or in a suitable appendix. In the FEA simulation where thermoacoustic effects taken into account? If so, details needs to be provided about the implementation. There is a problematic lack of information regarding the FEA implementation of the system. All the equations in Appendix D are a mess!

Author Response

The authors present an interesting bit of work. I like it generally but I have some comments and I would like a reply to each one of them before I can suggest it for publication. 

Thank you for finding this work interesting. Below, we are going to accurately answer all your comments and suggestions.

Point 1: The authors should discuss optical methods (e.g. photogrammetry) for volumetric measurements. 

Response 1: We accidentally missed these methods in the introduction, but we had mentioned them later in the text, where seed densities were measured. We now added some text in the introduction at new lines 33-41.

” In this case noninvasive methods should be used.

One of these methods is optical. Here, projections or stereoscopic images of an object at different angles with subsequent 3D-reconstruction allow calculating the object volume [3]. This method can be fast, but requires significant computational resources, as well as multiple cameras or a single camera moving relative to the object. Currently, we are using this method for phenotyping small plant seeds [4,5], but this technique is generally used for objects from micro- to macro-scale [6–8]. However, it does not work for all geometries or for objects with rough surfaces [5], since concave structures may not or only partly be visible for cameras.”

We also added an overview of an additional modality of the acoustic volumeters (new lines 53-57):

“There is also another type of acoustic volumeters based on Helmholtz acoustic resonance. In this method the volume of a cavity defines a resonating sound pitch, which can be determined by a microphone after the excitation of acoustic waves with different frequencies using a loudspeaker at the entrance of the cavity. This technique is used to measure, e.g., tank filling by a liquid [21–23] or combustion chamber volumes [24].”

Point 2: Line 109 – Can the authors explain the validity of the assumption that the frequency response is flat. This is an important aspect as many of the presented results show a frequency dependence which may be accounted for with the speaker response. A FRF of the transducer used in the study should be presented in the appendix.

Response 2: In fact, the sentence “Here we assume a flat response, i.e., R)=1.” (new line 137) contradicts the previous statement and does not have a crucial influence on the final results. It has now been deleted. Eqs. (6a), (6b), and (7) were also changed introducing R) and RS)in their numerators, respectively. And the sentences

“where RS) is the frequency response of the loudspeaker when the object is present inside the chamber. Supposing that R) = RS), which holds in case of operating far from internal resonances of the loudspeaker 5, the”

were added starting from new line 159. We also added some explanation starting from line 165:

“It should be noted that by working at frequencies where the assumption   is questionable, an additional microphone, here 8, should be used to control the frequency response of the loudspeaker 5 by correcting the applied signal.”

Additionally, we modified all formulas in Appendix C and added two lines in the Nomenclature section (new lines 430-431):

R)             Frequency response of the loudspeaker when the chamber is empty

 RS)           Frequency response of the loudspeaker when an object is present inside the chamber”

We also added new Appendix D, which include the FRF of the transducer.

Point 3: Pictures of the experimental set-up should be included.

Response 3: We added Figure 1b and a related caption. Additionally, the following text was added in new line 115: “The measurement setup is presented in Figure 1b.”

Point 4: Line 138 – The same number of significant figures should be used.

Response 4: We changed the text accordingly to ““… different diameters of 1.000, 1.588, 2.500, 4.000, 5.000, 6.000, 7.000, 8.000, and 9.000 (mm; ±0.011 mm) …”.

Point 5: Figure 2. The inserted diagrams of different set-ups should be larger. There is enough room.

Response 5: We increased the size of the inserted diagrams.

Point 6: Equation 12. At the right hand side there is a corrupted symbol. Equation 14. Same as above.

Response 6: We apologize for the troubles which were caused by the conversion of the .docx file to the .pdf. All equations are repaired now.

Point 7: Line 219. Some details of the FEA implementation should be added (in an appendix possibly). Mesh type, element size, time domain or frequency domain etc.

Response 7: The following explanation was added starting from new line 259:

“Laminar isothermal flow, as well as 2D-axisymmetric analysis, were selected to reduce the simulation time. The time domain simulations were performed using a triangular type of mesh automatically generated by COMSOL. Mesh size depended on the length and radius of the simulated tube as well as on the chamber volume, resulting in a number of elements ranging from 3000 to 20000. We performed Mesh Convergence Analysis for several simulated structures to confirm the mesh size selection.”

Additionally to that, the version of COMSOL used was added in line 256.

Point 8: Corrupted symbols. These are all throughout the paper and all of them need to be addressed.

Response 8: We are very sorry about all the inconvenience we caused. Please see our response to Point 6.

Point 9: Line 243. Why is the frequency range to specific? More detail is needed on exactly which devices were used for both the transmitting and receiving signals.

Response 9: The originally selected frequency range was from 20 Hz and up to 6000 Hz using 45 points on a logarithmic scale. The data from 3000 Hz and up to 6000 Hz was not shown to improve data readability (clarity), as these points are not influencing the following discussion of the obtained data. Moreover, we did not perform simulation for the frequencies higher than 3000 Hz to decrease their running time.

It should be noted that we did not skip the frequency range of the second resonance peak, as it had no significant impact on the measurements.

“both types of earphones were used in this work” was added in new line 89.

“The sensor with an inner volume of 6.5 mm radius and 3 mm length was used for these measurements.” was added in new lines 284-285.

More detail about devices used for both the transmitting and the receiving signals we added in Figure 1b. The following new sentence was also included in new lines 122-123: “To control the multifunction DAQ device the LabView (National Instruments, USA) -based program was developed and used.”

Point 10: Line 246 talks about resonance effects of the system but no data is presented to confirm this. This data needs to be included here or in a suitable appendix.

Response 10: We added this information in an additional Appendix D.

Additionally, we modified the sentence in lines 291-293, which now reads: “The frequencies between 700 and 1000 Hz were skipped during the measurements due to possible instabilities of the sensor operation close to its main resonance frequency of ~800 Hz (see Appendix D).”

Point 11: In the FEA simulation where thermoacoustic effects taken into account? If so, details needs to be provided about the implementation. There is a problematic lack of information regarding the FEA implementation of the system.

Response 11: Thank you for this point, but no, we did not take the thermoacoustic effect into account within this study. For the frequencies used, the sound wavelength is at least 4 times higher than the largest distance between two points inside the measured chamber. And, inside the measured tubes/pores we supposed to have isothermal behaviour, which is justified by the small distances to the walls (see Figure 10).

We added the following text starting from new line 178:

“Preferably, the working frequency should not exceed a frequency with corresponding sound wavelength less than 4 times larger than the largest distance within the chamber. This will avoid uncertainties caused by acoustic effects, such as standing waves, diffraction, etc.”

Point 12: All the equations in Appendix D are a mess!

Response 12: We apologize. Please see our response to Point 6.

Reviewer 2 Report

This is an interesting study, well suited to Sensors as a novel approach to estimation of volume.  The authors have comprehensively explained their approach, with sufficient theory, and provided appropriate experimental validation. The writing is clear and terse.

The only constructive criticisms I can offer are (a) the Introduction could cover expected limitations to the technique, (b) the experimental work could provide some detail on the detection limits of the technique. How would the technique apply to, say, estimation of pore space in a small volume of soil? 

Trivial items are failure to use a space between value and unit ('10mm') and use of the colloquial 'lets' (li 199).

Author Response

This is an interesting study, well suited to Sensors as a novel approach to estimation of volume.  The authors have comprehensively explained their approach, with sufficient theory, and provided appropriate experimental validation. The writing is clear and terse.

Thank you very much for your kind review of our work.

Point 1: The Introduction could cover expected limitations to the technique.

Response 1: We modified the introduction accordingly. In new line 60 we added the following sentence:

“Acoustic volumeters have few intrinsic limitations. Object size is generally limited only by volumeter size. And the detection of closed voids inside the object requires highly sophisticated modalities, such as magnetic resonance imaging [25] or x-ray computed tomography [26], instead of acoustic volumeters.”

Point 2: The experimental work could provide some detail on the detection limits of the technique. How would the technique apply to, say, estimation of pore space in a small volume of soil?

Response 2: The second paragraph of Section 4.1.1. now provides some limits of the technique. Additionally, we added the following text in new line 178:

“Preferably, the working frequency should not exceed a frequency with a corresponding sound wavelength less than 4 times larger than the largest distance within the chamber. This will avoid uncertainties caused by acoustic effects, such as standing waves, diffraction, etc.”.

We also added the following text in new line 392:

“The measurements at different frequencies demonstrate that the selection of a narrow frequency operation range may lead to inaccurate volume estimation when using an acoustic volumeter. For objects with pores on the surface, a measurement at higher frequencies would result in volume overestimation since pores might be not visible then. Ideally, only working at infinitely low frequencies (like gas pycnometry) would allow to obtain the `true´ volume of an object. In practice, measurement at low frequencies, e.g., around 232 Hz, may provide already quite reliable results for e.g., plant seeds.”

We were also thinking about the estimation of pore space in a volume of soil. A further study might be related to that. But within the scope of this work, where we introduce the method, we did not want to overload the story.

Point 3: Trivial items are failure to use a space between value and unit ('10mm') and use of the colloquial 'lets' (li 199).

Response 3: We made the suggested changes throughout the text.

Reviewer 3 Report

The article is sufficiently novel and interesting to warrant publication, the paper fits well to the topic of journal and provides interesting results. However the paper has several limitations.

Abstract: The abstract is expected to include a brief digest of the research, that is, new methods, results, concepts, and conclusions only. The abstract needs to be more focused and achievements needs mentioned clearly. At the moment abstract is more like an patent information than abstract of science article. Please add some information from the conclusion (quantifications).

Introduction based on old references and this is weak point of this paper.  I personally feel that this part of paper is not concise enough from a reader’s perspective. Introduction must provide a comprehensive critical review of recent developments in a specific area or theme that is within the scope of the journal (measurement), not only a list of published studies or a bibliometric one. Introduction is expected to have an extensive literature review followed by an in-depth and critical analysis of the state of the art. If you avoided reference overkill/run-on, i.e. do not use more than 3 references per sentence. If you need to use more, make sure you state the key relevant idea of each reference. References section should be extensive about information connecting with measurement and instrumentation. I suggest add information to better describe what other researchers have done in this area. Additionally, Authors do not write their paper in the context of Instrumentation and Measurement. Authors should present their work properly within the existing metrology literature; i.e., papers published in the metrology and measurement journals, and compare their work with these latest related Measurement work. This comparison should be done analytically (in the introduction or related work section), experimentally (in the performance evaluation section), or both.

The strengths and limitations of the applied approach should be clearly identified for the readers of the paper.

The discussion is shallow and needs more details, the observations and future trends. This chapter should be connected with others published papers.

Some of the bullet points on the conclusion are simplistic;  Please try to emphasize your novelty, put some quantifications, and comment on the limitations. This is a very common way to write conclusions for a learned academic journal. The conclusions should highlight the novelty and advance in understanding presented in the work.

Author Response

The article is sufficiently novel and interesting to warrant publication, the paper fits well to the topic of journal and provides interesting results. However the paper has several limitations.

Point 1: Abstract: The abstract is expected to include a brief digest of the research, that is, new methods, results, concepts, and conclusions only. The abstract needs to be more focused and achievements needs mentioned clearly. At the moment abstract is more like an patent information than abstract of science article. Please add some information from the conclusion (quantifications).

Response 1: Thank you for this point. While we agree about the general purpose of an abstract, we want to point out that the main message of this work is in the novel design of the sensor, not so much in quantitative results. Yet, in order to cover your point, we added some quantitative information to the abstract, which now reads:

“We introduce a novel technique to measure volumes of any shaped objects based on acoustic components. The focus is on small objects with rough surfaces, such as plant seeds. The method allows measurement of object volumes more than 1000 times smaller than the volume of the sensor chamber with both high precision and high accuracy. The method is fast, noninvasive, and easy to produce and to use. The measurement principle is supported by theory describing the behavior of the measured data for objects of known volumes in a range of 1 to 800 µL. In addition to single-frequency, we present frequency-dependent measurements that provide supplementary information about pores on the surface of a measured object such as the total volume of pores and in the case of cylindrical pores their average radius-to-length ratio. We demonstrate the usefulness of the method for seed phenotyping by measuring the volume of irregularly shaped seeds and showing the ability to “look” under the husk and inside pores, which allows to assess the true density of seeds.” 

Point 2: Introduction based on old references and this is weak point of this paper.  I personally feel that this part of paper is not concise enough from a reader’s perspective. Introduction must provide a comprehensive critical review of recent developments in a specific area or theme that is within the scope of the journal (measurement), not only a list of published studies or a bibliometric one. Introduction is expected to have an extensive literature review followed by an in-depth and critical analysis of the state of the art. If you avoided reference overkill/run-on, i.e. do not use more than 3 references per sentence. If you need to use more, make sure you state the key relevant idea of each reference. References section should be extensive about information connecting with measurement and instrumentation. I suggest add information to better describe what other researchers have done in this area. Additionally, Authors do not write their paper in the context of Instrumentation and Measurement. Authors should present their work properly within the existing metrology literature; i.e., papers published in the metrology and measurement journals, and compare their work with these latest related Measurement work. This comparison should be done analytically (in the introduction or related work section), experimentally (in the performance evaluation section), or both.

Response 2: We apologize for this misunderstanding. Our novel technique is not meant as a contribution to metrology, but as a practical (i.e., simple to build and operate) tool which extends the use of acoustic volumetry in several ways: enabling measurement of small object volumes (e.g. from around 1 µL to several mL) and objects with rough surfaces which are investigated by measurements over a large range of frequencies. To our knowledge, none of these core aspects of our work is covered by the existing metrology literature, which mainly deals with the measurement of mass standards. This is the reason why we refrained from a detailed comparison of our work with existing metrology literature in the introduction as well as in the discussion section.

To clarify this and other points we substantially modified the introduction. We added information about acoustic volumeters based on Helmholtz acoustic resonance in new lines 53-57. We added intrinsic limitations of acoustic volumeters in new lines 60-63. We made changes in new lines 71-79, which now read:

“Moreover, these volumeters were designed to operate at a narrow frequency range and were used predominantly for the measurements of objects with smooth surfaces and volumes larger than 1 mL, such as mass standards [17,20].

Here, we present a novel acoustic volumeter [27], which was developed to overcome the mentioned issues. It can be easily manufactured and used, and enables accurate measurements of any shaped objects with volumes from 1 µL to the mL range. We also show that, by operating the acoustic volumeter at different frequencies, additional surface information can be obtained such as surface porosity of an object. The theory behind the new device will be presented and possible applications will be discussed.”

Point 3: The strengths and limitations of the applied approach should be clearly identified for the readers of the paper.

Response 3: We clarified the strengths and limitations of the approach in the abstract (see the response to point 1), in the introduction (see the response to point 2) and in the conclusions (see the response to point 5).

In the results and discussion section, we added the following text in new lines 178-181:

“Preferably, the working frequency should not exceed a frequency with a corresponding sound wavelength less than 4 times larger than the largest distance within the chamber. This will avoid uncertainties caused by acoustic effects, such as standing waves, diffraction, etc.”

We added the following text in new lines 279-280:

“Internal porosity of the object, however, is not accessible with this approach and would require tomographic methods [25,26].”

We added the following text in new lines 392-398:

“The measurements at different frequencies demonstrate that the selection of a narrow frequency operation range may lead to inaccurate volume estimation when using an acoustic volumeter. For objects with pores on the surface, a measurement at higher frequencies would result in volume overestimation since pores might be not visible then. Ideally, only working at infinitely low frequencies (like gas pycnometry) would allow to obtain the `true´ volume of an object. In practice, measurement at low frequencies, e.g., around 232 Hz, may provide already quite reliable results for e.g., plant seeds.”

Point 4: The discussion is shallow and needs more details, the observations and future trends. This chapter should be connected with others published papers.

Response 4: In the discussion, we added the text mentioned in response 3. However, we did not include a comparison with metrology literature into the discussion for reasons explained in response 2.

Point 5: Some of the bullet points on the conclusion are simplistic;  Please try to emphasize your novelty, put some quantifications, and comment on the limitations. This is a very common way to write conclusions for a learned academic journal. The conclusions should highlight the novelty and advance in understanding presented in the work.

Response 5: We rewrote the conclusion, emphasizing the novelties. It now reads:

“We presented a simple acoustic volumeter, described its measuring principle analytically and validated it by measuring objects of known and well-defined size. The system is applicable to objects of any shape and has some novel features. (1) It has a large range of operation, starting at volumes of around 1 µL up to several mL, which can be further scaled up or scaled down. (2) The sensor is rather simple to construct, consisting basically of two tubes, two electromechanical transducers and some electronics. (3) The acoustic volumeter is easy to operate. A test object is put on a flat plate and then enclosed by pulling the sensor over the plate. (4) Volume detection is fast. A few seconds are needed to gather enough signal points to estimate object volumes more than 1000 times smaller than the volume of the sensor chamber with both high precision and high accuracy. (5) The acoustic volumeter can be operated at different frequencies between, e.g., ~20 and ~3000 Hz. This allows to estimate the `true´ volume of an object with rough surface, and to assess the total volume of pores on the surface of an object as well as the relation of the pore dimensions rp/Lp. When plant seeds were investigated, different volumes measured at different frequencies indicated porous surfaces. This can be used to gather seed properties which were so far not approachable.”

Round 2

Reviewer 1 Report

I thank the authors for addressing each of my points and giving excellent comments. I think the paper is very well put together with the minor changes and I'm more than happy to suggest it for publication with no further modifications.